# The performance model of logistic distribution centers: Quality function deployment based on the Best-Worst Method

Le Ngo Ngoc Thu[1], Long Van Hoang[2], Quynh Manh Doan[3,4], Nguyen Tan Huynh[3,4]*

**1** Institute of Postgraduate Studies, HUTECH University, Ho Chi Minh City, Vietnam, **2** Faculty of Management, Ho Chi Minh City University of Law, Ho Chi Minh City, Vietnam, **3** Applied Economics Research Group, Dong Nai Technology University, Bien Hoa City, Vietnam, **4** Faculty of Economics - Management, Dong Nai Technology University, Bien Hoa City, Vietnam

* huynhtannguyen@dntu.edu.vn

**Data Availability Statement:** All relevant data are within the manuscript and its Supporting information files.

## Abstract

This paper aims to evaluate the performance of logistic distribution centers (LDCs) by Quality Function Deployment (QFD) based on the Best-Worst Method (BWM). The originalities of this paper include: (1) exploring five constructs with 20 customer requirement attributes (CRAs) for LDCs' operations based on the SERQUAL model, (2) using the QFD model to identify five primary divisions and 18 corresponding service technical requirements (STRs) for LDCs' operations, (3) recognizing the top five STRs should be prioritized for the allocation of limited resources from House of Quality (HoQ), including cargo order (7.15%), value-added activities (6.68%), document preparation (6.31%), consolidating and assembling (6.10%), and document management (6.00%), (4) applying the Best-Worst Method (BWM) to estimate CRAs' relative weight. The proposed research model can provide a methodological reference to the relevant literature in association with logistics operations and multiple-criteria decision analysis (MCDA).

## 1. Introduction

Logistical distribution centers (LDCs) are argued to be a strategic cornerstone within the intricate tapestry of modern supply chain management, especially in the era of e-commerce [1–3]. Also, LDCs are storage facilities where goods from manufacturers or suppliers are stored, undergone scrutiny—meticulous inspection to ensure quality and quantity meet exacting standards [4], and then delivered to customers [5]. Nevertheless, their roles have extended far beyond storage in the modern time, including order fulfillment [6], inventory management [1], cross-docking [7], quality control [8], transportation and shipping [9], returns processing [10], value-added services [11]. Accordingly, improving LDCs' performance has been receiving much attention from academics and practitioners.

Until now, the pertinent research in terms of boosting LDCs' operating performance often relates to customer perspectives. Such research works try to identify customer requirement attributes (CRAs), which typically refer to specific characteristics or qualities that are essential

**Funding:** The author(s) received no specific funding for this work.

**Competing interests:** The authors have declared that no competing interests exist.

**Abbreviations:** LDCs, Logistic distribution centers; QFD, Quality Function Deployment; BWM, Best-Worst Method; CRAs, Customer requirement attributes; STRs, Service technical requirements; HoQ, House of Quality; WHATs, The voice of the customers; HOWs, Quality characteristics; CI, Consistency index; CR, Consistency ratio; DMs, Decision makers; WMS, Warehouse Management System; TMS, Transport Management System; CRM, Customer Relationship Management; ERP, Enterprise Resource Planning; FMCG, Fast-Moving Consumer Goods.

for meeting the needs and expectations of customers when it comes to a particular service or product [12, 13]. Relevant studies have postulated that LDCs' customers pay much attention to some CRAs, such as high product quality [14, 15], quick customer inquiry response [16], low shipping errors [17, 18], accurate delivery, and convenient order procedure [19, 20]. The above-mentioned CRAs are defined as the "what" issues of LDCs, which refer to the identification and analysis of customer needs, expectations, and requirements [21, 22]. Nonetheless, the strategies to satisfy CRAs are associated with STRs of LDCs, which can be defined as "how" issues [23]. Therefore, identifying STRs not only helps LDCs enhance service quality efficiently but also determines which service operations and corresponding divisions in their organization should be improved.

In addition, assessing the operating performance of LDCs is closely associated with the MCDA process, which is defined as making decisions in the presence of many criteria (or objectives). The systematic review by Basílio, Pereira [24] argued that the five most-popular methods of MCDA consist of AHP (Analytic Hierarchy Process), TOPSIS (Technique for Order of Preference by Similarity to Ideal Solution), VIKOR (VIseKriterijumska Optimizacija I Kompromisno Resenje), PROMETHEE (Preference Ranking Organization Method for Enrichment Evaluation), ANP (Analytic network process). More specifically, TOPSIS and VIKOR belong to multi-attribute utility and value theories. Meanwhile, AHP, PROMETHEE, and ANP are classified as outranking methods, which are based on pairwise comparisons among alternatives with respect to each criterion. Moreover, Hsu, Huang [25] posited that the extensive workload required for pairwise comparisons can complicate the application of MCDA, especially in the case of many criteria or alternatives involved, thus reducing consistency in these comparisons. Therefore, BWM can overcome this drawback since it requires fewer pairwise comparisons than AHP, PROMETHEE, and ANP. Besides, it uses integers from 1, 2,. . ., and 9 to do pairwise comparisons, thereby improving its practical applications. To the best of our knowledge, no single prior literature adopts BWM in assessing the performance of LDCs. It can be said that the current paper can contribute to the application of BWM towards logistics operation research.

This paper aims to evaluate the performance of LDCs by adopting the QFD model based on BWM. In doing so, the article first figures out LDCs' CRAs from their customers' perspective. Once CRAs are established, the next step is to determine the technical and engineering specifications (i.e., STRs) needed to meet customer requirements. Then, the BWM-based QFD model is adopted to translate CRAs into STRs via House of Quality (HoQ). Consequently, STRs are quantified and ranked for improvement policies. As an empirical study, LDCs in Vietnam (the VN-LDCs case) are investigated to verify the proposed research model.

This paper consists of seven sections. Section 2 gives a review of the relevant literature. Research methods are described in Section 3. The results of empirical research are given in Section 4. Sensitive analysis and comparison of STRs' ranking are shown in Sections 5 and 6, respectively. Finally, the overall conclusions and future directions are presented in Section 7.

## 2. Literature review

### 2.1. Overview of quality function deployment

Quality Function Deployment (QFD) is a comprehensive and structured methodology used in the field of manufacturing [26], quality management [23], product development [21], and risk reduction [27]. It originates from Japan and has gained widespread recognition for its effectiveness in translating customer needs (i.e., CRAs) into specific engineering and design requirements (i.e., STRs). It has been posited that QFD is a systematic process that facilitates the alignment of customer requirements with product or service characteristics [23], ultimately

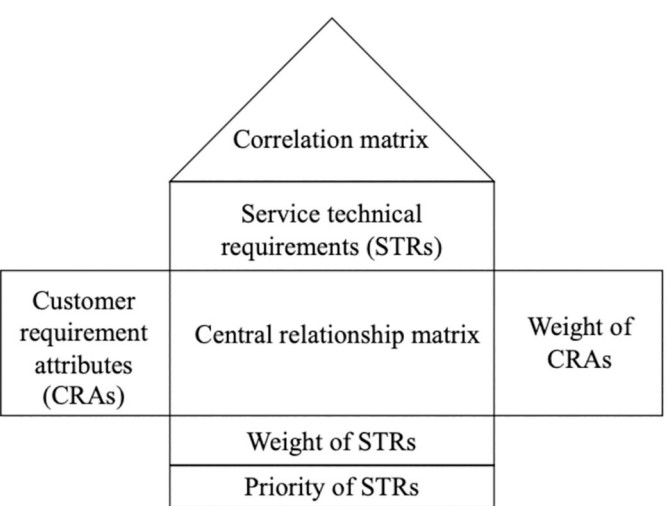

**Fig 1. The visualization of the house of quality.**

leading to improved customer satisfaction and the delivery of high-quality offerings [28]. QFD emphasizes the importance of capturing the "voice of the customer" by diligently collecting and analyzing customer feedback and preferences [21, 26]. Aydin, Seker [29] and Paltayian, Georgiou [30] argued that this initial step is quite pivotal, as it forms the foundation upon which the entire QFD process is built.

The central tool of QFD is the "House of Quality" matrix that visually represents the relationships between the voice of the customers (WHATs) and the quality characteristics (HOWs), as seen in Fig 1. For LDCs' operations, the WHATs are the site requirements from the customers' perspective while the HOWs are the evaluation criteria that LDC operators must satisfy. On top of that, the triangular roof of HoQ is the correlation matrix displaying the interrelationships among individual STRs [31] and identifying how the design requirements interact with each other [32].

The roots of QFD can be traced back to Japan in the 1940s and 1950s, primarily in the manufacturing sector. Yet, its application has been extended to some service industries, such as public bus transportation [31], electronic supply chains [32], entrepreneurial education services [26], retailing industries [29], ocean freight forwarders [33], airfreight [34], and port risk management [35].

## 2.2. Best-worst method

As mentioned earlier, Some MCDM methods (i.e., AHP, ANP, PROMETHEE) suffer from various drawbacks, such as redundant pairwise comparisons and the lack of consistency. To fill this gap, Rezaei [36] introduced a new MCDA method, named BWM, to carry out pairwise comparisons in a structured way. In essence, the pairwise comparison of BWM relies on the reference criterion, which is defined as the best element or the worst one. In this sense, the most preferred criterion ($C_B$) and the least preferred one ($C_W$) are pre-determined as proxy for references for all remaining criteria. Aftermaths, decision-makers only need to carry out pairwise comparisons based on references in the decision-making procedure. As a result, we form ($n-1$) times of pairwise comparisons regarding the best criterion, and ($n$-2) time of pairwise comparisons regarding the worst criterion. In other words, the number of pairwise

comparisons in BWM is $2n - 3$ in total, contrary to AHP, ANP, and PROMETHEE needing $n(n-1)/2$ pairwise comparisons.

Since its introduction, BWM has attracted much attention from academics and practitioners. The research work published by Rezaei [36] has become one of the most cited research in Omega since 2014 [37]. Up till now, BWM has been used in multiple fields, such as strategic property management [38], sustainable supplier selection and order allocation [39], airport selection [40], meat supply chain [37], lithium battery plant allocation [41], land consolidation projects [42], evaluation of ecological governance in the Yellow River basin [43], etc. Also, Mi, Tang [37] predicted the increasing trend of utilization and adaption of BWM in the coming future.

Although BWM is so famous in the MCDA field, its application raises much concern that should be addressed. First, according to Liang, Brunelli [44], the challenge of BWM is the lack of consistency thresholds to determine the reliability of the results. It is widely admitted that the consistency ratio of 10% is acceptable. Yet, the logic of this threshold is still questionable. Secondly, Rezaei [45] uses the linearized form of the original non-linear model to solve multi-optimality in the BWM. Still, the usage of the linearized techniques to analyze the multi-optimality of BWM is challengeable for the many, especially in the case of many criteria and alternatives involved. Wang, Fu [43] also have the same idea. Accordingly, developing software packages can help simplify the calculation of BWM, and in turn, assist its utilization in solving MCDA issues.

## 2.3. The performance of LDCs

The rise of e-commerce has dramatically reshaped the retail landscape, leading to shifts in the logistics and supply chain sectors; accordingly, LDCs have evolved to play a central role in supporting the growth and demands of e-commerce. As noted earlier, most relevant literature discusses the performance of LDCs from their customers' view. Below is the summary of some typical research, as shown in Table 1.

Kuo, Dunn [6] assessed performance measurement in distribution centers by exploratory and descriptive analysis to determine "best practices" and opportunities for improvement. According to research, measurement systems in LDCs typically fall into six categories: finance, operations, quality, safety, personnel, and customer satisfaction. Nozick and Turnquist [10] developed an analysis procedure for the location of LDCs that integrates facility costs, inventory costs, transportation costs, and service responsiveness. It is argued that there is a fundamental trade-off between customer responsiveness and costs when designing LDCs. Chen [1] applied the fuzzy set theory to the external performance evaluation of LCDs under an uncertain environment. Through personal interviews, the study explores six criteria with 19 sub-criteria measuring the performance of LCDs, including efficiency, customers, stockouts, delivery, order, and personnel. Besides, empirical results show that efficiency is the most important criterion for the performance measurement of LCDs. Ross and Droge [8] assessed the performance of LCDs using DEA modeling. Some criteria are used in this paper, comprising fleet size (number of vehicles), experience (average years of experience of direct labor), mean order throughput time in days, sales volume, and total revenues. Results identify the best performance LCDs for determining tactics and strategies to improve LDCs' performance.

Voss, Calantone [9] examined how front-line employee performance and interdepartmental customer orientation affected the service, supply chain, and financial performance of the U.S. LDCs. It is illustrated that the performance of the front-line employee directly affects the service rendered to the external customer and supply chain performance. This finding is relatively consistent with Ross and Droge [8]. Lu and Yang [11] evaluated key logistics capabilities for

**Table 1. Summary of the relevant literature.**

| Source | Research objectives | Methods | Main findings |
|---|---|---|---|
| Kuo et al. (1999) | Investigating the measurement systems used in LDCs. | Exploratory and descriptive analysis | • Most LDCs measure productivity or efficiency.<br>• The greatest opportunity for improvement appears to be in the area of customer satisfaction measures.<br>• Safety was the most consistently measured category. |
| Nozick and Turnquist (2001) | Identifying locations for LDCs | The fixed-charge facility location model | • Some factors involve: facility costs, inventory costs, transportation costs and service responsiveness.<br>• The trade-off between customer responsiveness and costs. |
| Chen (2002) | Evaluating external performance for LDCs | Fuzzy AHP-like method | • Six criteria with 19 sub-criteria measuring the performance of LDCs.<br>• Efficiency is the most important criterion for the performance measurement of LDCs. |
| Ross and Droge (2002) | Assessing the performance of LCDs | DEA modeling | • Some factors involve fleet size (number of vehicles), experience (average years of experience of direct labor), mean order throughput time in days, sales volume, and total. revenues.<br>• Exploring the relevance of internal comparative advantage among LDCs. |
| Voss et al. (2005) | Examining the relationship between front-line employee performance and LDCs' efficiency | Ordinary least squares | • The front-line employee directly affects the service rendered to the external customer and supply chain performance.<br>• High levels of service in each internal transaction lead to greater potential for delivery of high levels of service to the external customer. |
| Lu and Yang (2010) | Evaluating key logistics capabilities for LDCs | SEM | • Primary criteria for LDCs' performance include customer response, innovation, economic scale, flexible operation, and logistics knowledge.<br>• Customer response capability is the most important criterion for the performance measurement of LDCs. |
| Shi et al. (2013) | Reconfiguring cross-docking LDCs in a Chinese—Japanese joint venture supply chain | Robust optimization | • Some performance metrics are used, including dwelling time, throughput, receiving doors, shipping doors, forklifts, conveyors, etc.<br>• Results can address what-if questions to optimize LDCs' performance. |
| Holzapfel et al. (2018) | Optimizing product allocation to different types of LDCs in retail logistics networks | The MIP model | • Some criteria are considered such as inbound and outbound transportation, warehouse operations, and in-store logistics.<br>• The proposed model can assist LDCs managers in making allocation decisions and enhancing competitiveness. |
| Keshavarz-Ghorabaee (2021) | Evaluating LDCs locations | SWARA II | • Some crucial decisive criteria are adopted: quality of service, proximity of customers and suppliers, environmental impacts, and operating costs.<br>• The proposed approach can be applied to a case of LDCs locations assessment. |

LDCs in Taiwan by the structural equation modeling (SEM). Based on exploratory factor analysis, primary criteria for LDCs' performance were explored, including customer response, innovation, economic scale, flexible operation, and logistics knowledge. Findings suggest that customer response capability is the most important criterion for the performance measurement of LDCs, followed by flexible operation, logistics knowledge, innovation, and economic scale capability.

Shi, Liu [12] proposed a hybrid modeling that consists of discrete-event system simulation, and robust optimization to reconfigure cross-docking LDCs in a Chinese—Japanese joint venture supply chain. Some performance metrics are used, including dwelling time, throughput, receiving doors, shipping doors, forklifts, conveyors, and threshold time allowed to dwell in a temporary storage area. Results can address what-if questions to optimize LDCs' performance. Holzapfel, Kuhn [19] optimized product allocation to different types of LDCs in retail logistics networks via the MIP model. Some criteria are considered such as inbound and outbound

transportation, warehouse operations, and in-store logistics. The proposed research model can assist LDCs managers in making allocation decisions and enhancing competitiveness in an ever tougher market environment for bricks-and-mortar retailers. Keshavarz-Ghorabaee [14] evaluated LDCs locations using a multi-expert subjective—objective decision-making approach. Some crucial decisive criteria are adopted in this article, such as quality of service, proximity of customers and suppliers, environmental impacts, and operating costs. The proposed approach can be applied to a case of distribution center locations assessment.

Although the performance of LDCs has been discussed thoroughly in the relevant research, two major literature gaps should be addressed.

First, it is argued that all of the pertinent research assesses LDCs' operating performance from customer perspectives. In doing so, they determine CRAs, which typically refer to the specific characteristics or criteria that define the quality and scope of a service. Nevertheless, how to translate CRAs into STRs to meet customers' needs is undocumented. In the present article, this gap is filled by using the QFD model.

Secondly, to estimate the relative weight of customer-related criteria, the prior studies adopt some classic methods, such as the fixed-charge facility model [10], AHP [1, 46], DEA [8], multivariate technique [9, 47], SEM [11], optimization [12], MIP [19], SWARA [14]. To the best of our knowledge, while BWM has been deployed extensively in supply chain management, its deployment to assess the performance of LDCs has never been documented. Hence, the current paper can contribute to the application of BWM towards the logistics operation research.

## 3. Research methods

### 3.1. Research framework

The research process in this article is shown in Fig 2. Based on the literature review and the operating features of LDCs, CRAs for their operations are first identified. Then, the Best-Worst Method (BWM) developed by Rezaei [36] is adopted to evaluate the weights of the CRAs from users' perspectives. It is argued that the application of MCDA in the prior research is to focus on some common tools, for instance, ANP, AHP [48], DEA [8, 49], SWARA [14]. Still, such tools rely on quite complicated algorithms while the questionnaire design is relatively perplexing. Therefore, to overcome this challenge, BWM measuring the relative weights

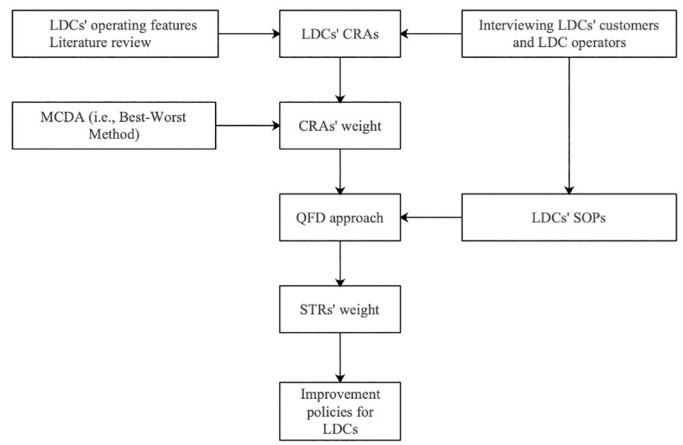

**Fig 2. Research framework.**

of the CRAs is adopted in this article. Next, the QFD model is established to translate CRAs into STRs. Finally, the weight of STRs might provide managerial information for LDCs' operators in improving their operational performance.

## 3.2. The structural configuration of CRAs

Based on the literature review and the operating features of LDCs, the structural configuration of CRAs is created, comprising five constructs in Layer 1 and 20 criteria in Layer 2 (Table 2), as follows:

Tangibility (TB): the appearance of physical facilities, equipment, personnel, and communication materials. For LDCs' operations, tangibility includes 4 criteria, such as the handling and transportation of products, convenient location, enough space to store cargo, and well-maintained and organized facility.

Reliability (RB): the ability of LDCs to consistently deliver accurate, promised services without error. It underscores the notion of dependability and trustworthiness in the service delivery process.

Responsiveness (RS): the willingness or readiness of employees to assist customers and provide prompt service. This dimension touches on the agility and adaptability of LDCs in meeting their customers' needs and addressing their concerns.

Empathy (EP): the capacity to affirm customers' feelings and indicate that LDCs can understand their concerns. Besides, this construct emphasizes the emotional and interpersonal interactions between LDCs and their customers.

Assurance (AS): the ability of LDCs to relay trust to their customers which is paramount in getting confidence in customers to stick with the organization as opposed to the competition.

**Table 2. CRAs' hierarchical structure.**

| Constructs | Criteria | Code | Sources |
|---|---|---|---|
| Tangibility (TB) | The handling and transportation of products | TB1 | Chen (2002), Ren et al. (2009), Abbasi et al. (2021) |
| | Convenient location | TB2 | Holzapfel et al. (2018), Lin and Lu (1999) |
| | Enough space to store cargoes | TB3 | Shi et al. (2013), Kilibarda et al. (2011) |
| | Well-maintained and organized facility | TB4 | Lu and Yang (2010), Aydin et al. (2023) |
| Reliability (RB) | On-time delivery | RB1 | Nozick and Turnquist (2001), Voss et al. (2005) |
| | Correctness of shipping orders | RB2 | Lizarelli et al. (2021), Ross and Droge (2002) |
| | Order Accuracy | RB3 | Experts' suggestion |
| | Safety and Security | RB4 | Miciak and Desmarais (2001), Keshavarz-Ghorabaee (2021), Nguyen et al. (2022) |
| Responsiveness (RS) | Order processing speed | RS1 | Liang et al. (2006), Chen (2002), |
| | Clear and timely communication with customers | RS2 | Sharma et al. (2008), Voss et al. (2005) |
| | Efficient and cost-effective transportation | RS3 | Ren et al. (2009), Abbasi et al. (2021) |
| | Inventory availability | RS4 | Liang et al. (2006), Lu and Yang (2010), Aydin et al. (2023) |
| Empathy (EP) | Understand and address customer concerns and issues | EP1 | Holzapfel et al. (2018), Aydin et al. (2023) |
| | Communicate the situation clearly, express sympathy for any inconvenience caused, and proactively offer solutions or alternatives | EP2 | Lin and Lu (1999), Kilibarda et al. (2011), Aydin et al. (2023) |
| | Be flexible in adapting operations to meet customer demands and expectations. | EP3 | Nozick and Turnquist (2001), Lin and Lu (1999) |
| | Understanding and addressing the unique needs of different customers. | EP4 | Voss et al. (2005), Nguyen et al. (2022) |
| Assurance (AS) | Complies with all relevant laws and regulations | AS1 | Experts' suggestion |
| | Offer responsive and effective customer support | AS2 | Miciak and Desmarais (2001), Abbasi et al. (2021) |
| | Establish a feedback loop with customers and stakeholders | AS3 | Chen (2002), Nguyen et al. (2022) |
| | Continuously monitor and evaluate processes | AS4 | Ren et al. (2009), Sharma et al. (2008) |

Moreover, assurance specifically deals with the knowledge and courtesy of employees and their ability to convey trust and confidence.

## 3.3. Research sample and questionnaire design

According to the Vietnam Ministry of Industry and Trade [50], there are 33 LDCs across Vietnam as of 2022. More specifically, the vast majority of these are located in the major cities and industrial provinces of Vietnam, such as HCM city, Hanoi, Danang, Dong Nai, Binh Duong, and Hai Phong. Some LDCs with a major operational scale include Gemadept, TBS Logistics, Vinalines Logistics, Saigon Newport, Transimex, DKSH Vietnam, Damco Vietnam, etc. To select a representative sample, this current study chose the four biggest LDCs, viz., Gemadept, JBS Logistics, Vinalines Logistics, and Saigon Newport for the empirical case. After that, the research team asked these LDCs to provide 10 major customers. As a result, we had a sample of 40 respondents for the survey.

Because of adopting BWM to determine CRAs' weight, referring to Rezaei [36], this paper designed the nine-point pairwise-comparison scale, as presented in Table 3, to survey 40 respondents, as mentioned above. Note that due to the highly professional nature of the questions in the questionnaire, all of the participants being surveyed were required to have adequate professional expertise in the logistics industry. From April to July 2023, we managed to gather information from such 40 experts. Nevertheless, 3 responses were discarded as they did not satisfy the consistency of BWM. Consequently, only 37 responses are valid for the next analysis. Respondents' background is also shown in Table 4.

Evidently, 62.2% of respondents hold managerial positions at their workplaces while 56.8% of them have working experience of more than 11 years in the distribution field. Additionally, 70.3% of respondents working at LDCs have an operating year of more than 15 years. Moreover, some main types of products of LDCs include pharmaceuticals and healthcare products, industrial and construction equipment, Fast-Moving Consumer Goods (FMCG), apparel and fashion, automotive parts and accessories, and electronics and telecommunications equipment.

## 3.4. The weight of CRAs

As mentioned previously, this paper adopts BWM to derive the priority weights of the criteria (i.e., CRAs). So, this section presents the main steps to conduct BWM [36, 51].

**Step 1**. Select a set of decision criteria (i.e., CRAs). Suppose that we have a set of criteria $C = (C_1, C_2, \ldots, C_j, \ldots, C_n)$ for decision-making. Where $j = 1, 2, \ldots, n$ is the number of the evaluated criteria.

**Table 3. Linguistic variables for the pairwise comparison and the consistency index.**

| Scale | Linguistic term | Consistency index (maxξ) |
|:---:|:---:|:---:|
| 1 | Equally Important (EI) | 0.00 |
| 2 | Intermediate of EI & MI (IEM) | 0.44 |
| 3 | Moderately Important (MI) | 1.00 |
| 4 | Intermediate of MI & I (IMI) | 1.63 |
| 5 | Important (I) | 2.30 |
| 6 | Intermediate of I & VI (IVI) | 3.00 |
| 7 | Very Important (VI) | 3.73 |
| 8 | Intermediate of VI & EXI (IEI) | 4.47 |
| 9 | Extremely Important (EXI) | 5.23 |

**Table 4. Respondents' background.**

| Characteristics | | Frequency | % |
|---|---|---|---|
| Gender | Male | 33 | 89.2 |
| | Female | 4 | 10.8 |
| Age in years | 25 ~ 30 | 5 | 13.5 |
| | 31 ~ 40 | 17 | 45.9 |
| | 41–40 | 9 | 24.3 |
| | Above 50 | 6 | 16.2 |
| Educational level in years | Undergraduate | 17 | 45.9 |
| | Master | 18 | 48.6 |
| | Ph.D | 2 | 5.4 |
| Working experience (years) | 5 ~ 10 | 16 | 43.2 |
| | 11 ~ 20 | 11 | 29.7 |
| | 21 ~ 30 | 6 | 16.2 |
| | Above 30 | 4 | 10.8 |
| Working position | Head of the Division | 12 | 32.4 |
| | Assisstant manager | 4 | 10.8 |
| | Vice manager | 7 | 18.9 |
| | Senior staff | 14 | 37.8 |
| Type of main products | Pharmaceuticals and Healthcare Products | 5 | 13.5 |
| | Industrial and Construction Equipment | 7 | 18.9 |
| | Fast-Moving Consumer Goods (FMCG) | 9 | 24.3 |
| | Apparel and Fashion | 7 | 18.9 |
| | Automotive parts and accessories | 4 | 10.8 |
| | Electronics and telecommunications equipment | 5 | 13.5 |
| Company age (years) | Under 15 | 11 | 29.7 |
| | 15 ~ 25 | 17 | 45.9 |
| | 26 ~ 35 | 4 | 10.8 |
| | Above 35 | 5 | 13.5 |
| Revenues (million USD) | Under 15 | 9 | 24.3 |
| | 16 ~ 25 | 12 | 32.4 |
| | 26 ~ 55 | 6 | 16.2 |
| | Above 55 | 10 | 27.0 |

**Step 2**. Establish a panel of decision-makers (DMs) $E = (E_1, E_2, \ldots, E_k, \ldots, E_h)$ for the decision-making process. And $k = 1, 2, \ldots, h$ is the number of DMs in the sample.

**Step 3**. Identify the best criteria ($C_B$) and the worst criteria ($C_W$). Note that no comparison judgment is carried out at this step. Note that each expert will determine the best and the worst criteria differently.

**Step 4**. Use integer numbers from 1 to 9 (Table 3), each DM rates the preference of the best criterion over all the other criteria. As a result, we obtain $A_B^k = \left( a_{B1}^k, a_{B2}^k, \ldots, a_{Bj}^k, \ldots, a_{Bn}^k \right)$ where $a_{Bj}^k$ is the preference of the best criterion $B$ over criterion $j$, which is judged by the $k^{th}$ expert. It is clear that $a_{BB}^k = 1$.

**Step 5**. Similarly, every DM rates the preference of all the other criteria over the worst criterion. As a result, we have $A_W^k = \left( a_{1W}^k, a_{2W}^k, \ldots, a_{jW}^k, \ldots, a_{nW}^k \right)$ where $a_{jW}^k$ is the preference of

the criterion $j$ over the worst criterion $W$, which is judged by the $k^{th}$ expert. It is clear that $a_{WW}^k = 1$.

**Step 6**. Find the individual optimal weights $W^k = \left( w_1^k, w_2^k, \ldots, w_j^k, \ldots, w_n^k \right)$. The optimal weight for the criteria will satisfy the condition, for each pair of $w_B^k / w_j^k$ and $w_j^k / w_W^k$, we have $w_B^k / w_j^k = a_{Bj}^k$ and $w_j^k / w_W^k = a_{jW}^k$ In doing so, we can find a solution to minimize the absolute maximum differences $\left| \frac{w_B^k}{w_j^k} - a_{Bj}^k \right|$ and $\left| \frac{w_j^k}{w_W^k} - a_{jW}^k \right|$ Then, a min-max optimization model is formulated as follows:

$$\min \max_j \left\{ \left| \frac{w_B^k}{w_j^k} - a_{Bj}^k \right|, \left| \frac{w_j^k}{w_W^k} - a_{jW}^{\mathbf{k}} \right| \right\}$$

S.t.

$$\sum_{j=1}^n w_j^k = 1$$

$$w_j^k \geq 0 \text{ for all } j$$

(1)

Model (1) can be transformed into Model (2):

$$\min \xi^k$$

S.t.

$$\left| \frac{w_B^k}{w_j^k} - a_{Bj}^k \right| \leq \xi^k \text{ for all } j$$

$$\left| \frac{w_j^k}{w_W^k} - a_{jW}^k \right| \leq \xi^k \text{ for all } j$$

$$\sum_{j=1}^n w_j^k = 1$$

$$w_j^k \geq 0 \text{ for all } j$$

(2)

**Step 7**. Check consistency for expert judgement using consistency ratio (CR):

$$CR = \frac{\xi^{k*}}{CI}$$

(3)

Where $\xi^{k*}$ is the objective value from Model (2) while $CI$ is the consistency index, as shown in the last field of Table 3.

**Step 8**. Determine the combined optimal weight for all judgements $W = (w_1, w_2, \ldots, w_j, \ldots, w_n)$. From Step 6, we have $h$ individual priority vectors ($w^k$) from $h$ DMs. Finally, such vectors are combined together to form the relative weight of criteria using the geometric mean, as

**Table 5. Weights of CRAs.**

| Layer 1: Constructs | Weight of Layer 1 | Layer 2: Criteria | Local weight of Layer 1 | Global weight of Layer 1 |
|---|---|---|---|---|
| Tangibility (TB) | 31.25 | TB1 | 29.98 | 9.37 |
| | | TB2 | 25.51 | 7.97 |
| | | TB3 | 26.81 | 8.38 |
| | | TB4 | 17.71 | 5.53 |
| Reliability (RB) | 27.97 | RB1 | 16.82 | 4.70 |
| | | RB2 | 28.07 | 7.85 |
| | | RB3 | 26.65 | 7.45 |
| | | RB4 | 28.46 | 7.96 |
| Responsiveness (RS) | 15.28 | RS1 | 23.72 | 3.62 |
| | | RS2 | 14.00 | 2.14 |
| | | RS3 | 29.55 | 4.51 |
| | | RS4 | 32.74 | 5.00 |
| Empathy (EP) | 17.79 | EP1 | 14.83 | 2.64 |
| | | EP2 | 29.60 | 5.27 |
| | | EP3 | 17.61 | 3.13 |
| | | EP4 | 37.95 | 6.75 |
| Assurance (AS) | 7.71 | AS1 | 15.18 | 1.17 |
| | | AS2 | 29.56 | 2.28 |
| | | AS3 | 16.63 | 1.28 |
| | | AS4 | 38.62 | 2.98 |

follows:

$$w_j^* = \left[\prod_{k=1}^{h} \left(w_j^{k*}\right)\right]^{\frac{1}{h}}, j = 1, 2, \cdots, n \tag{4}$$

Applying Steps (1) to (8), the priority weight of CRAs in the VN-LDCs case can be determined as shown in Table 5.

### 3.5. The QFD model

**3.5.1. The identification of STRs.** To develop the QFD model, the first thing is to identify LDCs' STRs to meet customers' requirements [21, 29]. From standard operating procedures (SOPs) and the organizational structure of 4 sampled LDCs, as presented in Section 3.3, interviewed experts reach a high consensus that LDCs' operation is associated with five primary divisions, including the Business Department, the Customer Department, the Customs Department, the Warehouse Department, and the Transport & Delivery Department. From that, 18 STRs are created, as exhibited in Fig 3, including 2 STRs (i.e., cargo order, cargo forwarding) from the Business Department, 4 STRs (i.e., customer complaint, order contract, document preparation, and shipment processing) from the Customer Department, 2 STRs (i.e., X-ray inspection and document checking) from the Custom Department, 6 STRs (i.e., cargo receiving and checking, sorting and putting away, consolidating and assembling, value-added activities, storage management, and shipment returns) from the Warehouse Department, and 4 STRs (i.e., delivery planning, document management, shipment loading, and delivery) from the Transport & Delivery Department.

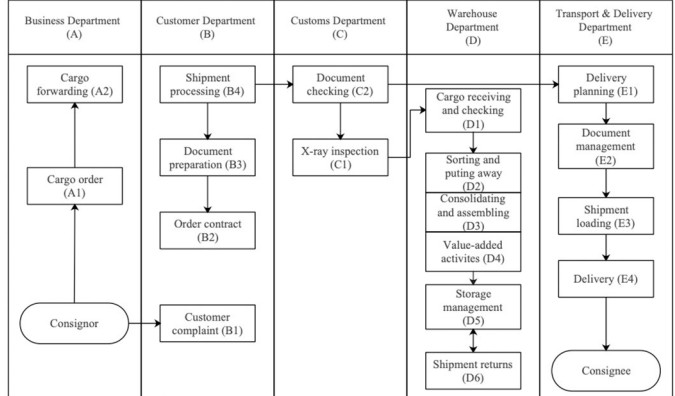

**Fig 3. The operating procedure of logistical distribution centers.**

**3.5.2. The central relationship matrix and correlation matrix.** To set up HoQ for translating CRAs into STRs, the paper calculates the central relationship matrix and the correlation matrix. More specifically, the central relationship matrix is measured from 0 (non-relationship) to 4 (very high relationship). Meanwhile, the correlation coefficient is limited between [0; 1]. Where the correlation coefficient of 0 means there is no linear correlation while the coefficient of 1 shows a perfect positive correlation among STRs. At last, to assign coefficients for such two matrices, the current research randomly selects seven industrial experts from the 4 sampled LDCs, whose profile is displayed in Table 6.

**3.5.3. STRs' weight.** Suppose that one HoQ is formed by $u$ CRAs and $v$ STRs. As such, HoQ is represented by the relationship matrix $T = [T_{ij}]_{u \times v}$ $(i = 1,2,\ldots,u; j = 1,2,\ldots,v)$ Let $T_{ij}^k(k = 1, 2, \cdots, h)$ be the relationship coefficient assigned by the $k^{th}$ expert. Then $T_{ij}$ is determined by the geometric mean:

$$T_{ij} = \left[ \prod_{k=1}^{h} T_{ij}^k \right]^{\frac{1}{h}} \tag{5}$$

Then, $T_{ij}$ is standardized by:

$$T_{ij}^v = \frac{T_{ij}}{\sum\limits_{j=1}^{v} T_{ij}} (i = 1, 2, \ldots, u; j = 1, 2, \ldots v) \tag{6}$$

**Table 6. Experts' background.**

| Experts | Working experience (years) | Working position | Field experience |
|---|---|---|---|
| 1 | 20 | Vice manager | Warehouse |
| 2 | 5 | Senior staff | Customer services |
| 3 | 9 | Vice manager | Business |
| 4 | 15 | Vice manager | Transport |
| 5 | 15 | Head of the Division | Transport and Customs |
| 6 | 5 | Head of the Division | Warehouse and customer services |
| 7 | 11 | Head of the Division | Business and warehouse |

**Table 7. The QFD results for STRs.**

| | | Business Dept. | | Customer Dept. | | | | Custom Dept. | | Warehouse Dept. | | | | | | Transport & Delivery Dept. | | | | Weights of CRAs |
|---|---|---|---|---|---|---|---|---|---|---|---|---|---|---|---|---|---|---|---|---|
| | | A1 | A2 | B1 | B2 | B3 | B4 | C1 | C2 | D1 | D2 | D3 | D4 | D5 | D6 | E1 | E2 | E3 | E4 | |
| Tangibility (TB) | TB1 | 0.06 | 0.09 | 0.03 | 0.06 | 0.03 | 0.03 | 0.04 | 0.10 | 0.05 | 0.08 | 0.08 | 0.05 | 0.07 | 0.03 | 0.03 | 0.06 | 0.06 | 0.05 | 9.37 |
| | TB2 | 0.07 | 0.04 | 0.07 | 0.06 | 0.04 | 0.05 | 0.07 | 0.03 | 0.05 | 0.04 | 0.07 | 0.06 | 0.04 | 0.07 | 0.07 | 0.06 | 0.07 | 0.03 | 7.97 |
| | TB3 | 0.04 | 0.04 | 0.08 | 0.07 | 0.09 | 0.05 | 0.03 | 0.08 | 0.08 | 0.05 | 0.02 | 0.07 | 0.08 | 0.03 | 0.07 | 0.03 | 0.06 | 0.03 | 8.38 |
| | TB4 | 0.04 | 0.05 | 0.08 | 0.03 | 0.04 | 0.05 | 0.05 | 0.03 | 0.05 | 0.09 | 0.08 | 0.09 | 0.07 | 0.06 | 0.03 | 0.07 | 0.05 | 0.06 | 5.53 |
| Reliability (RB) | RB1 | 0.08 | 0.04 | 0.04 | 0.06 | 0.05 | 0.07 | 0.06 | 0.06 | 0.07 | 0.07 | 0.02 | 0.04 | 0.06 | 0.07 | 0.04 | 0.05 | 0.05 | 0.06 | 4.70 |
| | RB2 | 0.03 | 0.05 | 0.02 | 0.04 | 0.07 | 0.05 | 0.06 | 0.07 | 0.07 | 0.07 | 0.08 | 0.07 | 0.03 | 0.07 | 0.05 | 0.02 | 0.07 | 0.06 | 7.85 |
| | RB3 | 0.07 | 0.04 | 0.04 | 0.05 | 0.08 | 0.05 | 0.04 | 0.04 | 0.07 | 0.10 | 0.05 | 0.03 | 0.04 | 0.04 | 0.09 | 0.08 | 0.06 | 0.03 | 7.45 |
| | RB4 | 0.05 | 0.06 | 0.04 | 0.07 | 0.07 | 0.08 | 0.04 | 0.04 | 0.06 | 0.06 | 0.02 | 0.05 | 0.08 | 0.08 | 0.03 | 0.03 | 0.05 | 0.09 | 7.96 |
| Responsiveness (RS) | RS1 | 0.04 | 0.07 | 0.07 | 0.07 | 0.07 | 0.05 | 0.05 | 0.07 | 0.03 | 0.06 | 0.06 | 0.04 | 0.06 | 0.08 | 0.06 | 0.03 | 0.03 | 0.05 | 3.62 |
| | RS2 | 0.04 | 0.04 | 0.06 | 0.06 | 0.07 | 0.07 | 0.05 | 0.07 | 0.03 | 0.08 | 0.07 | 0.06 | 0.08 | 0.03 | 0.04 | 0.04 | 0.05 | 0.07 | 2.14 |
| | RS3 | 0.05 | 0.03 | 0.07 | 0.06 | 0.08 | 0.04 | 0.07 | 0.04 | 0.03 | 0.03 | 0.04 | 0.04 | 0.04 | 0.10 | 0.08 | 0.08 | 0.09 | 0.03 | 4.51 |
| | RS4 | 0.06 | 0.08 | 0.04 | 0.06 | 0.04 | 0.04 | 0.07 | 0.08 | 0.05 | 0.08 | 0.08 | 0.05 | 0.04 | 0.04 | 0.03 | 0.07 | 0.08 | 0.02 | 5.00 |
| Empathy (EP) | EP1 | 0.07 | 0.08 | 0.04 | 0.03 | 0.06 | 0.06 | 0.08 | 0.03 | 0.05 | 0.06 | 0.07 | 0.05 | 0.02 | 0.08 | 0.05 | 0.08 | 0.07 | 0.02 | 2.64 |
| | EP2 | 0.06 | 0.08 | 0.05 | 0.04 | 0.09 | 0.08 | 0.05 | 0.08 | 0.06 | 0.05 | 0.02 | 0.04 | 0.05 | 0.02 | 0.03 | 0.06 | 0.08 | 0.05 | 5.27 |
| | EP3 | 0.08 | 0.04 | 0.08 | 0.07 | 0.03 | 0.02 | 0.05 | 0.05 | 0.08 | 0.04 | 0.07 | 0.03 | 0.06 | 0.03 | 0.07 | 0.06 | 0.06 | 0.07 | 3.13 |
| | EP4 | 0.06 | 0.05 | 0.05 | 0.07 | 0.07 | 0.04 | 0.05 | 0.07 | 0.07 | 0.05 | 0.06 | 0.07 | 0.04 | 0.05 | 0.06 | 0.06 | 0.06 | 0.03 | 6.75 |
| Assurance (AS) | AS1 | 0.05 | 0.06 | 0.05 | 0.04 | 0.04 | 0.07 | 0.06 | 0.07 | 0.08 | 0.03 | 0.05 | 0.03 | 0.07 | 0.04 | 0.07 | 0.05 | 0.06 | 0.07 | 1.17 |
| | AS2 | 0.03 | 0.03 | 0.05 | 0.04 | 0.07 | 0.06 | 0.06 | 0.07 | 0.05 | 0.04 | 0.07 | 0.03 | 0.07 | 0.05 | 0.03 | 0.06 | 0.07 | 0.08 | 2.28 |
| | AS3 | 0.07 | 0.06 | 0.08 | 0.06 | 0.08 | 0.05 | 0.06 | 0.05 | 0.03 | 0.07 | 0.04 | 0.05 | 0.08 | 0.04 | 0.04 | 0.05 | 0.04 | 0.05 | 1.28 |
| | AS4 | 0.03 | 0.06 | 0.06 | 0.07 | 0.03 | 0.05 | 0.08 | 0.06 | 0.03 | 0.04 | 0.08 | 0.05 | 0.05 | 0.09 | 0.04 | 0.06 | 0.03 | 0.08 | 2.98 |
| Normalized weights of STRs | | 7.15 | 5.58 | 5.14 | 5.06 | 6.31 | 4.89 | 4.59 | 4.74 | 5.02 | 5.90 | 6.10 | 6.68 | 5.39 | 5.83 | 5.53 | 6.00 | 4.23 | 5.86 | |
| Weight of Dept. | | 6.36 | | 5.35 | | | | 4.66 | | 5.82 | | | | | | 5.41 | | | | |

Where $T_{ij}^v$ is the contribution level of the $i^{th}$ CRAs to the $j^{th}$ STRs. The $T_{ij}^v$ values are calculated and shown in Table 7.

Next, let $w_i^{CRA}(i = 1, 2, \cdots, u)$ be the weight of CRAs, whose values are presented in the last field of Table 7. Keep in mind that these values are obtained from the last column of Table 5. Then, the weight of STRs might be attained by:

$$M_j^{STR} = \sum_{i=1}^{u} \left( w_i^{CRA} \times T_{ij}^v \right) (i = 1, 2, \ldots, u; j = 1, 2, \ldots, v) \tag{7}$$

Further, let $N = [N_{ij}]_{v \times v}$ $(i, j = 1, 2, \ldots, v)$ is the correlation matrix. Suppose that $N_{ij}^k$ the correlation coefficient between the $i^{th}$ STR and $j^{th}$ STR, judged by the $k^{th}$ expert. It is clear that $N_{ij} = 1$ for all $i = j$. Then $N_{ij}$ $(i, j = 1, 2, \ldots, v)$ is determined by Formula (8) and their results build the roof of HoQ, as illustrated in Fig 4.

$$N_{ij} = \left[ \prod_{k=1}^{h} N_{ij}^k \right]^{\frac{1}{h}} \tag{8}$$

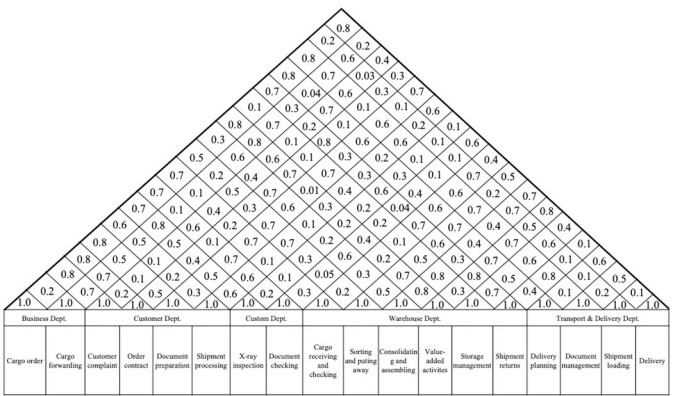

**Fig 4. The roof of house of quality.**

Now, STRs' weight, as computed by Formula (7), can then be revised as follows:

$$W_j^{STR} = M_i^{STR} + \sum_{i \neq j} \left( N_{ij} \times M_i^{STR} \right) \tag{9}$$

Finally, $W_j^{STR}$ is normalized to figure out its ranking by:

$$\omega_j^{STR} = \frac{W_j^{STR}}{\sum_{j=1}^{v} W_j^{STR}} \times 100 \tag{10}$$

The weight of STRs and corresponding divisions for the VN-LDCs are shown in the second-last and last row of Table 7, respectively.

## 4. Discussion

Table 5 demonstrates that LDCs' customers pay most of their attention to Tangibility (31.25%) and Reliability (27.97%), followed by Empathy (17.79%), Responsiveness (15.28%), and Assurance (7.71%). More specifically, the top five CRAs in Layer 2 consist of the *handling and transportation of products* (9.37%), *enough space to store cargoes* (8.38%), *convenient location* (7.97%), *safety and security* (7.96%), and the *correctness of shipping orders* (7.85%). Besides, as seen in Table 7, the top five STRs need to be improved to meet customer requirements, including *cargo order* (7.15%), *value-added activities* (6.68%), *document preparation* (6.31%), *consolidating and assembling* (6.10%), and *document management* (6.00%).

From the above-mentioned empirical results for the VN-LDCs, the research team carried out a post-interview with some managers of sampled LDCs. Consequently, managerial policies were recommended to enhance LDCs' operating performance, as follows:

### Build and expand larger storage space to better the efficacy of cargo consolidation

To boost STRs, such as *value-added activities* and *consolidating and assembling,* larger storage space will be paramount of importance. It is argued that for LDCs' operations, cargo consolidation is to package, bundle, and combine many separate cargo items into one large shipment for distribution to the same end location [52]. It also helps accelerate delivery time, reduces

operational costs, and develops long-term collaboration with partners, thus increasing transportation efficiency [17, 19]. Doing consolidation and assembling requires looking at cargoes in more detail. So, this kind of task requires enough storage space and obliges workers to work carefully and concentrated. Moreover, to improve more efficacy of cargo consolidation, LDCs' operators should invest more money to build and expand storage space. This recommendation is relatively consistent with Shi, Liu [12]. Moreover, improving the value-added activities of LDCs can lead to increased efficiency, cost savings, and enhanced customer satisfaction. Experts suggested raise the turnover rate of the storage space by slotting optimization. To fulfill that, LDCs regularly review and optimize the placement of products within the distribution center to minimize travel time for order pickers and maximize space utilization.

### Improving working abilities for front-line staff through periodic training courses

According to the empirical results, *document preparation* (6.31%) and *document management* (6.00%) are two STRs needing to be reallocated scarce resources. Currently, document management and file processing in LDCs predominantly involve weekly work schedules for the entire company and a variety of specific tasks regarding trucks and routes arrangement and coordination, customer contact, and responses, especially document preparation for customs clearance. These areas of work require highly professional, cooperative, and experienced. A small mistake may lead to a big problem, such as shipment delay [13], or wrong information in the bill of lading [52]. Besides, responding precisely and quickly to customer inquiries is an essential requirement for LDCs' operations. Thus, experts suggested that front-line employees should be trained every three to four months to improve their professional knowledge and soft skills.

### Upgrading management information system

The cargo order (7.15%) is the number-one priority for LDCs to improve. To achieve that, interviewed experts advised that LDCs should set up a management information system (MIS) to collect, process, store, and distribute information. They also added that an effective MIS may support customer-service staff in responding exactly and quickly to customers' requirements and complaints. Many studies have argued that the integration of MIS in the operational process could optimize the performance capacity of distribution centers [53] and prevent shipping accidents [54]. In addition, to properly manage cargo orders for the smooth operation of LDCs, experts suggested integrate different software systems, for instance, Warehouse Management System (WMS), Transport Management System (TMS), Customer Relationship Management (CRM), and Enterprise Resource Planning (ERP).

## 5. Sensitivity analysis

The paper uses subjective evaluation data to determine the weights of the indicators, which have a significant impact on the final ranking of STRs. Accordingly, the current paper performs sensitivity analysis on the indicator weights to enhance the robustness of the research, as what has been done by Yang, Chen [55], Finger and Lima-Junior [56], and Du, Liu [57]. To handle that, this paper first randomly divides 37 respondents, as presented in Section 3.3, into various subgroups as shown in Table 8. Then, the current study calculates the weight of CRAs and STRs and ranks STRs, as displayed in Table 9.

It is evident that the final ranking of STRs has a relatively high consistency with various scenarios. Particularly, some higher-ranking STRs (i.e., A1, B3, D3, D4, and E2) are almost the same under different scenarios. Nonetheless, few STRs have slight changes in priority ranking,

**Table 8. Some scenarios in sensitive analysis.**

| Scenario | Subgroups |
|---|---|
| Scenario 1 | Subgroup 1: E10, E4, E14, E29, E31, E15, E21, E36, E18, E22, E30, E17, E26<br>Subgroup 2: E25, E28, E5, E27, E2, E33, E32, E13, E9, E37, E20, E16<br>Subgroup 3: E3, E24, E19, E12, E8, E7, E6, E35, E1, E23, E34, E11 |
| Scenario 2 | Subgroup 1: E13, E2, E16, E10, E1, E12, E20, E25, E9, E14<br>Subgroup 2: E19, E35, E34, E24, E11, E8, E7, E27, E23<br>Subgroup 3: E33, E3, E32, E15, E5, E31, E36, E30, E37<br>Subgroup 4: E17, E22, E6, E26, E28, E21, E29, E4, E18 |
| Scenario 3 | Subgroup 1: E5, E27, E14, E6, E15, E16, E33, E35<br>Subgroup 2: E8, E24, E23, E34, E3, E22, E37, E17<br>Subgroup 3: E25, E19, E28, E30, E2, E18, E11<br>Subgroup 4: E10, E21, E1, E29, E36, E4, E26<br>Subgroup 5: E31, E13, E9, E12, E7, E2,0 E32 |

for example, B1 in scenario 2, and D1, E1, A2 in scenario 3. To sum up, sensitive analysis demonstrates that the proposed QFD method has good robustness.

# 6. Comparisons of STRs' ranking among the different QFD model

To investigate further insight into the features of the proposed research method, the paper compares it with some MCDA-based QFD methods, such as AHP, ANP, TOPSIS, and GRA. To achieve this, we invite five representative experts from the research sample to evaluate the importance ratings of CRAs and the relationships between CRAs and STRs. Consequently, comparison results of STRs' ranking among the different QFD models are shown in Table 10 and Fig 5. It is demonstrated that there are discernible disparities among the five methods in ranking STRs. This is mainly because of the different algorithms of the methods. More specifically, while AHP and ANP use pairwise comparison to assess criteria, TOPSIS determines the best and the worst alternative for each criterion. Moreover, GRA requires decision-makers to

**Table 9. Ranking of STRs under different scenarios.**

| | A1 | A2 | B1 | B2 | B3 | B4 | C1 | C2 | D1 | D2 | D3 | D4 | D5 | D6 | E1 | E2 | E3 | E4 |
|---|---|---|---|---|---|---|---|---|---|---|---|---|---|---|---|---|---|---|
| Current scenario | 1 | 9 | 12 | 13 | 3 | 15 | 17 | 16 | 14 | 6 | 4 | 2 | 11 | 8 | 10 | 5 | 18 | 7 |
| Scenario 1 | 2 | 8 | 12 | 13 | 3 | 14 | 17 | 16 | 15 | 7 | 4 | 1 | 11 | 6 | 10 | 5 | 18 | 9 |
| Scenario 2 | 1 | 8 | 14 | 10 | 2 | 16 | 15 | 17 | 11 | 6 | 4 | 3 | 12 | 7 | 13 | 5 | 18 | 9 |
| Scenario 3 | 2 | 5 | 11 | 16 | 3 | 14 | 18 | 15 | 10 | 9 | 7 | 1 | 12 | 8 | 13 | 4 | 17 | 6 |

**Table 10. Ranking of STRs under the different QFD models.**

| | A1 | A2 | B1 | B2 | B3 | B4 | C1 | C2 | D1 | D2 | D3 | D4 | D5 | D6 | E1 | E2 | E3 | E4 |
|---|---|---|---|---|---|---|---|---|---|---|---|---|---|---|---|---|---|---|
| Proposed QFD | 1 | 9 | 12 | 13 | 3 | 15 | 17 | 16 | 14 | 6 | 4 | 2 | 11 | 8 | 10 | 5 | 18 | 7 |
| AHP-based QFD | 1 | 8 | 14 | 13 | 2 | 17 | 18 | 16 | 12 | 7 | 5 | 3 | 10 | 4 | 11 | 6 | 15 | 9 |
| ANP-based QFD | 3 | 8 | 13 | 12 | 5 | 15 | 16 | 17 | 10 | 9 | 1 | 2 | 11 | 6 | 14 | 7 | 18 | 4 |
| TOPSIS-based QFD | 1 | 8 | 14 | 18 | 2 | 17 | 11 | 16 | 15 | 12 | 6 | 7 | 9 | 5 | 10 | 3 | 13 | 4 |
| GRA-based QFD | 3 | 10 | 14 | 13 | 5 | 11 | 12 | 15 | 17 | 16 | 1 | 7 | 9 | 4 | 8 | 2 | 18 | 6 |

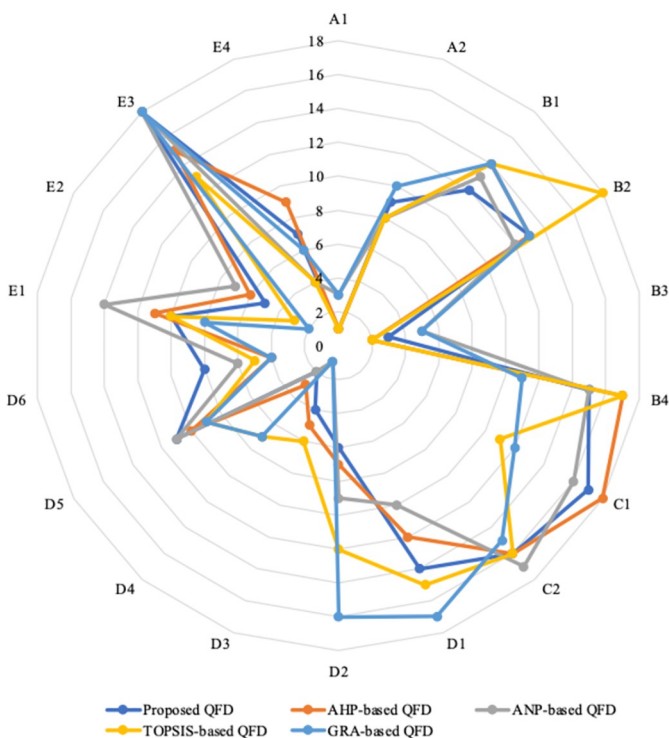

**Fig 5. Ranking of STRs under different QFD models.**

determine preference sequences and the dynamic distinguishing coefficient (ξ). Needless to say, selecting such parameters might significantly influence the final results.

In addition, the present paper adopts the cosine distance (*CD*) [58] to measure how close the ranking of STRs from the proposed QFD and the MCDA-based QFD is.

$$CD = 1 - \frac{\sum_{i=1}^{n} A_i B_i}{\sqrt{\sum_{i=1}^{n} A_i^2} \times \sqrt{\sum_{i=1}^{n} B_i^2}} \tag{11}$$

where $A_i$ and $B_i$ are the $i^{th}$ components of vectors *A* and *B*, respectively. The distance score between the proposed QFD and some MCDA-based QFD models (i.e., AHP, ANP, TOPSIS, and GRA) is 0.01, 0.02, 0.04, and 0.06, respectively. Evidently, the priority ranking of STRs from our proposed QFD and the AHP-based QFD is roughly in the same direction. Nevertheless, AHP requires more pairwise comparisons than our proposed model.

## 7. Conclusion

LDCs are crucial nodes in the supply chain and help facilitate the storage, processing, and distribution of goods. They primarily serve the locations of receiving products, storing them, and then redistributing them to retailers, wholesalers, or sometimes directly to consumers. Accordingly, enhancing the performance of LDCs is of paramount critical for a variety of reasons, such as spanning operational efficiency, cost savings, customer satisfaction, error reduction, working risk mitigation, etc. Nevertheless, to optimize the performance of LDCs, most of the

relevant research is based solely on customers' views while a few investigate LDCs' service operation attributes (i.e., STRs). To fill the literature gap, this paper aims at evaluating the performance of LDCs by the QFD model based on BWM. Some contributions can be addressed, as follows:

Theoretically, customer requirements and expectations (i.e., CRAs) in the context of LDCs can be translated into specific product or service features (i.e., STRs) via the QFD approach. As a result, this process assures the close alignment between finished products and services and customer preferences. In the current paper, the SERQUAL model identifies CRAs for LDCs' operation, which embraces five dimensions with 20 attributes. This is also the foundation for LDCs' operators in building service management requirements to satisfy their customers' requirements. Next, thanks to expert interview, the present research argues that LDCs' operation is associated with five primary divisions and 18 corresponding STRs. Besides, HoQ for the VN-LDCs case recognizes the top five STRs that should be prioritized for the allocation of limited resources, including *cargo order* (7.15%), *value-added activities* (6.68%), *document preparation* (6.31%), *consolidating and assembling* (6.10%), and *document management* (6.00%). It is practically illustrated that the QDF model cannot only be applicable to further investigate CRAs for a specific operation of LDCs' performance (i.e., customer services, business affairs, etc.), but also helpful in the design process and product development of various distribution systems (i.e., electric power distribution, gas distribution systems, agricultural product distribution, etc.).

Moreover, the application of BWM in estimating CRAs' relative weight can provide a methodological reference to solve MCDA issues. According to Rezaei [36] and Hasan, Ashraf [51], BWM requires fewer comparison pairs than other pairwise comparison methods (i.e., AHP, ANP). For instance, BWM only needs $(2n\text{-}3)$ pairwise comparisons, while AHP necessitates $n(n\text{-}1)/2$ ones. Accordingly, BWM can be easily applied in many real-world cases in terms of evaluating and prioritizing different options based on MCDA.

Finally, some limitations should be addressed for further studies. It has been postulated that LDCs are complex systems, thereby difficult to obtain precise measurements or where measurements are subject to various sources of variability [59]. Additionally, experts' judgment is often subjective, uncertain, and imprecise [60, 61]. Therefore, this paper suggests that future research should adopt fuzzy QFD and fuzzy BWM to model and control complex systems with uncertain parameters. Besides, the case in the paper involves a multi-person collaborative decision-making problem. It is recommended further studies should use group decision-making methods, especially considering the influence of different experts' weights on the decision outcome.

## Supporting information

**S1 File.**
(PDF)

## Acknowledgments

The authors would like to thank colleagues for very thoughtful reviews and critical comments, which have led to significant improvements to the early versions of the manuscript.

## Author Contributions

**Conceptualization:** Nguyen Tan Huynh.

**Data curation:** Le Ngo Ngoc Thu, Long Van Hoang, Nguyen Tan Huynh.

**Formal analysis:** Nguyen Tan Huynh.

**Investigation:** Quynh Manh Doan.

**Validation:** Quynh Manh Doan.

**Visualization:** Long Van Hoang, Quynh Manh Doan.

**Writing – review & editing:** Le Ngo Ngoc Thu, Long Van Hoang.

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
