## [Decision Letter · Decision Letter 0]

25 Oct 2023

PONE-D-23-31891

The performance model of logistic distribution centers: Quality Function Deployment based on the Best-Worst Method

PLOS ONE

Dear Dr. Huynh,

Thank you for submitting your manuscript to PLOS ONE. After careful consideration, we feel that it has merit but does not fully meet PLOS ONE’s publication criteria as it currently stands. Therefore, we invite you to submit a revised version of the manuscript that addresses the points raised during the review process. In the revised manuscript, the authors should  specifiy more clearly their work's contribution to the literature. 

We look forward to receiving your revised manuscript.

Kind regards,

Jin Liu

Academic Editor

PLOS ONE

3. PLOS requires an ORCID iD for the corresponding author in Editorial Manager on papers submitted after December 6th, 2016. Please ensure that you have an ORCID iD and that it is validated in Editorial Manager. To do this, go to ‘Update my Information’ (in the upper left-hand corner of the main menu), and click on the Fetch/Validate link next to the ORCID field. This will take you to the ORCID site and allow you to create a new iD or authenticate a pre-existing iD in Editorial Manager. Please see the following video for instructions on linking an ORCID iD to your Editorial Manager account: https://www.youtube.com/watch?v=_xcclfuvtxQ"".

4. We note that Figure 3 in your submission contain [map/satellite] images which may be copyrighted. All PLOS content is published under the Creative Commons Attribution License (CC BY 4.0), which means that the manuscript, images, and Supporting Information files will be freely available online, and any third party is permitted to access, download, copy, distribute, and use these materials in any way, even commercially, with proper attribution. For these reasons, we cannot publish previously copyrighted maps or satellite images created using proprietary data, such as Google software (Google Maps, Street View, and Earth). For more information, see our copyright guidelines: http://journals.plos.org/plosone/s/licenses-and-copyright.

a. You may seek permission from the original copyright holder of Figure 3 to publish the content specifically under the CC BY 4.0 license.  

Additional Editor Comments:

We note that one or more reviewers has recommended that you cite specific previously published works. As always, we recommend that you please review and evaluate the requested works to determine whether they are relevant and should be cited. It is not a requirement to cite these works. We appreciate your attention to this request.

Reviewers' comments:

Reviewer's Responses to Questions

**Comments to the Author**

1. Is the manuscript technically sound, and do the data support the conclusions?

Reviewer #1: Partly

Reviewer #2: Partly

Reviewer #3: Yes

2. Has the statistical analysis been performed appropriately and rigorously? 

Reviewer #1: Yes

Reviewer #2: N/A

Reviewer #3: Yes

3. Have the authors made all data underlying the findings in their manuscript fully available?

Reviewer #1: No

Reviewer #2: Yes

Reviewer #3: Yes

4. Is the manuscript presented in an intelligible fashion and written in standard English?

Reviewer #1: Yes

Reviewer #2: Yes

Reviewer #3: Yes

5. Review Comments to the Author

Reviewer #1: The performance model of logistic distribution centers: Quality Function Deployment

based on the Best-Worst Method

PLOS ONE

PONE-D-23-31891

The paper aims to evaluate the performance of logistic distribution centers (LDCs) by Quality Function Deployment (QFD) based on the Best-Worst Method (BWM). Some critical concerns are mentioned below.

In abstract, we cannot recognize the novelties and contributions of the paper.

-What are the research gaps in the literature?

-What are your arguments to prefer BWM instead of other weighting methods, such as AHP, FUCOM, LMAW, LBWA, etc. Discuss BWM’s advantages over other methods mentioned.

-Another deficiency of the paper is the BWM literature. You should add a new sub-section in LR about recent BWM papers in PLOS ONE and other respectful journals. For instance, Exploring factors affecting airport selection during the COVID-19 pandemic from air cargo carriers’ perspective through the triangular fuzzy Dombi-Bonferroni BWM methodology; Assessment and risk mitigation on halal meat supply chain using fuzzy best-worst method (BWM) and risk mitigation number (RMN); The Suitability-Feasibility-Acceptability Strategy Integrated with Bayesian BWM-MARCOS Methods to Determine the Optimal Lithium Battery Plant Located in South America; Neighborhood selection for a newcomer via a novel BWM-based revised MAIRCA integrated model: a case from the Coquimbo-La Serena conurbation, Chile; Prioritization of land consolidation projects using the multi-criteria Best-Worst Method: a case study from Poland.

-Check the whole file regarding English language and fix some grammatical mistakes

Reviewer #2: The authors of this paper conducted a study on the performance evaluation of LDCs by integrating the BWM and QFD methods. The overall structure and content of this work are acceptable. However, there are some comments and suggestions for the authors to consider when improving the quality of your work. Please see the attachment for details of my comments.

Reviewer #3: Thank you for this contribution. This is an interesting and timely manuscript. A good piece of work has been done, and I recommend its acceptance. The authors need to update the literature, especially in recent years. Please review the below references;

Elhegazy, H., Ebid, A., Mahdi, I., Haggag, S., & Abdul-Rashied, I. (2021). Implementing QFD in decision making for selecting the optimal structural system for buildings. Construction Innovation, 21(2), 345-360. doi:10.1108/CI-12-2019-0149

Elhegazy, H., Mahmoud, W., Eid, M., & Khairy, N. (2022). Quality Function Deployment Framework for Selecting Optimal Greenhouse Microclimate Control System. Journal of Industrial Integration and Management. doi:10.1142/S2424862222500105

6. PLOS authors have the option to publish the peer review history of their article (what does this mean?). If published, this will include your full peer review and any attached files.

Reviewer #1: No

Reviewer #2: No

Reviewer #3: No

---

## [Author Response · Author response to Decision Letter 0]

12 Jan 2024

Dear Academic Editor and Reviewers,

I hope this letter finds you well. I am writing to submit the revised version of my manuscript, "The performance model of logistic distribution centers: Quality Function Deployment based on the Best-Worst Method," which was originally submitted on October 1st 2023. I appreciate the careful review of the manuscript by the reviewers and the valuable feedback provided. I would like to express my gratitude for the constructive comments and suggestions made by the reviewers. Their insights have significantly contributed to the improvement of the manuscript. In response to the reviewers' comments, we have made the following key revisions:

Responses to Reviewer #1:

The paper aims to evaluate the performance of logistic distribution centers (LDCs) by Quality Function Deployment (QFD) based on the Best-Worst Method (BWM). Some critical concerns are mentioned below.

Comment# 1. In abstract, we cannot recognize the novelties and contributions of the paper.

Answer: Thanks so much for your comments. We already revised the abstract to present some originalities of the article (Seeing Rows 23-30 in red texts).

Comment# 2. What are the research gaps in the literature?

Answer: Thanks so much for your comments. We already revised the literature review section to present some research gaps (Seeing Rows 180-193 in red texts).

Comment# 3. What are your arguments to prefer BWM instead of other weighting methods, such as AHP, FUCOM, LMAW, LBWA, etc. Discuss BWM’s advantages over other methods mentioned.

Answer: Thanks so much for your comments. We already revised the introduction section to present why BWM is used in our paper. Besides, BWM’s advantages over other methods are discussed in this section (Seeing Rows 59-74 in red texts).

Comment# 4. Another deficiency of the paper is the BWM literature. You should add a new sub-section in LR about recent BWM papers in PLOS ONE and other respectful journals. For instance, Exploring factors affecting airport selection during the COVID-19 pandemic from air cargo carriers’ perspective through the triangular fuzzy Dombi-Bonferroni BWM methodology; Assessment and risk mitigation on halal meat supply chain using fuzzy best-worst method (BWM) and risk mitigation number (RMN); The Suitability-Feasibility-Acceptability Strategy Integrated with Bayesian BWM-MARCOS Methods to Determine the Optimal Lithium Battery Plant Located in South America; Neighborhood selection for a newcomer via a novel BWM-based revised MAIRCA integrated model: a case from the Coquimbo-La Serena conurbation, Chile; Prioritization of land consolidation projects using the multi-criteria Best-Worst Method: a case study from Poland.

Answer: Thanks so much for your comments. We already inserted one sub-section “2.2. Best-Worst method” to provide an overview of BWM. Besides, your suggested studies are valuable sources of reference in this sub-section (Seeing Rows 111-136 in red texts).

Comment# 5. Check the whole file regarding English language and fix some grammatical mistakes.

Answer: Thanks so much for your comments. We had one English native speaker review the manuscript and correct any grammatical errors.

Responses to Reviewer #2: 

The authors of this paper conducted a study on the performance evaluation of LDCs by integrating the BWM and QFD methods. The overall structure and content of this work are acceptable. However, there are some comments and suggestions for the authors to consider when improving the quality of your work. Please see the attachment for details of my comments.

The authors of this paper conducted a study on the performance evaluation of LDCs by integrating the BWM and QFD methods. The overall structure and content of this work are acceptable. However, the following suggestions are provided for the authors to consider when improving the quality of your work:

Comment# 1. The research background of the paper does not clearly explain why the QFD and BWM methods are necessary and how they are relevant to the research problem. Where do these methods align with the study?

Answer: Thanks so much for your comments. We already revised the introduction section to explain why the QFD and BWM methods are necessary in our study and how they are relevant to the evaluation of LDCs’ performance (Seeing Rows 46-74 in green and red texts). Moreover, we present where these methods align with the study in the section “3.1. Research framework” (Seeing Rows 196-205 in green texts).

Comment# 2. The literature review section needs to be rewritten. Currently, it is simply a list without explaining the connections between the literature and why these specific ones were chosen. As we all know there is a vast amount of research on the performance evaluation of LDCs, so we recommend the authors present and compare the literature in a table format. Additionally, we did not see any analysis of the existing literature, particularly the shortcomings of previous studies and how this paper intends to address them.

Answer: Thanks so much for your comments. We already revised the literature section to address your concerns. More specifically, we inserted one section to discuss the advantages and challenges of adopting BWM (Seeing Rows 111-136 in red texts). Additionally, we already inserted two paragraphs in the literature section to present some research gaps that were fulfilled in this manuscript (Seeing Rows 180-192 in red texts). Further, we presented and compared the previous literature in a table format (Seeing Table 1 in Row 571 in green texts).

Comment# 3. What is the logical framework of the CRA system constructed in this paper? It appears that the authors used the SERVQUAL model, but what is the relationship or difference between it and the existing evaluation indicators?

Answer: Thanks so much for your comments. To suggest strategies to improve LDCs’ performance, this paper started with identifying customer requirements (i.e., CRAs) on the basis of the SERQUAL model and the operating features of LDCs. It has been argued that the SERQUAL model is exceedingly useful in figuring out customer requirements in the service industry, including five primary constructs (i.e., Tangibility, Reliability, Responsiveness, Empathy, and Assurance). Yet, these constructs’ observed items are referred to the relevant literature regarding LDCs.

Comment# 4. The methods used in the paper are mature methods proposed by others, and we are concerned about the originality or novelty of the paper itself. The authors should summarize and emphasize the innovative aspects of this study.

Answer: Thanks so much for your comments. the originality or novelty of the paper is summarized in the conclusion section (Seeing Rows 415-434 in green texts). On top of that, we already revised the abstract to present some originalities of the article (Seeing Rows 23-30 in red texts).

Comment# 5. The paper does not provide a clear description of how to transition from CRAs to STRs. It seems that STRs were derived based on expert opinions without utilizing the HOQ tool.

Answer: Thanks so much for your comments. We would like to explain how to translate from CRAs to STRs, as follows:

First of all, we explored CRAs from the literature review and the operating features of LDCs. Then, BWM was used to measure CRAs’ relative weight with the sample size of 21 respondents. Afterwards, we identified STRs from LDCs’ standard operating procedures (SOPs) and experts’ consultations (as shown in Figure 3 of the manuscript). Next, to set up HoQ for translating CRAs into STRs, the paper calculated the central relationship matrix and the correlation matrix by surveying 7 experts. After that, the weight of STRs was determined by Formulas (5) – (10). Finally, we ranked STRs’ relative weight to prioritize the resource allocation.

Comment# 6. The case in the paper involves a multi-person collaborative decision-making problem. We recommend the author to use group decision-making methods, especially considering the influence of different experts' weights on the decision outcome.

Answer: Thanks so much for your comments. We really appreciate your suggestion. Yet, the locations of LDCs for the empirical case are so far. Accordingly, it will not be easy for us to run a roundtable discussion for group decision-making. Thus, we declared this point as a research limitation in our manuscript to accept your criticism (Seeing Rows 439-442 in green texts).

Comment# 7. The weights of the indicators have a significant impact on the final ranking of LDCs. The paper uses subjective evaluation data for this, so to enhance the robustness of the research, we suggest the author perform sensitivity analysis on the indicator weights. The following papers related to QFD and MCDMs may provide insights into this issue and should be referenced. (a) A Hesitant Fuzzy Linguistic QFD Approach for Formulating Sustainable Supplier Development Programs doi:10.1016/j.ijpe.2022.108428; (b) Large-scale group decision-making for prioritizing engineering characteristics in quality function deployment under comparative linguistic environment. doi:10.1016/j.asoc.2022.109359, (c) Enhancing Quality Function Deployment Through the Integration of Rough Set and Ordinal Priority Approach: A Case Study in Electric Vehicle Manufacturing etc.

Answer: Thanks so much for your comments. We already inserted one section to present a sensitive analysis (Seeing Rows 371-384 in green and red texts).

Comment# 8. The paper lacks a comparison with other quantitative research methods, which prevents it from demonstrating the superiority of this study. The case study only proves the feasibility of the research methods used in this paper.

Answer: Thanks so much for your comments. We already inserted one section to compare STRs’ ranking from the different QFD models (Seeing Rows 385-404 in green and red texts).

Comment# 9. The paper contains numerous grammar and spelling mistakes. For example, the LCD in section 2.2 should be LDC and the MWM in section 3.1 should be BWM. We recommend the author have a native English speaker carefully review and improve the writing.

Answer: Thanks so much for your comments. We had one English native speaker review the manuscript and correct any grammatical errors. 

Responses to Reviewer #3: Thank you for this contribution. This is an interesting and timely manuscript. A good piece of work has been done, and I recommend its acceptance. The authors need to update the literature, especially in recent years. 

Please review the below references:

Elhegazy, H., Ebid, A., Mahdi, I., Haggag, S., & Abdul-Rashied, I. (2021). Implementing QFD in decision making for selecting the optimal structural system for buildings. Construction Innovation, 21(2), 345-360. doi:10.1108/CI-12-2019-0149

Elhegazy, H., Mahmoud, W., Eid, M., & Khairy, N. (2022). Quality Function Deployment Framework for Selecting Optimal Greenhouse Microclimate Control System. Journal of Industrial Integration and Management. doi:10.1142/S2424862222500105.

Answer: Thanks so much for your comments. We revised the manuscript by incorporating some suggested research (Seeing Rows 46-58 in green texts). 

Responses to Academic Editor: Thanks so much for your suggestion. We already deleted Figure 3 from our manuscripts.

Once again, thank you for your dedication to advancing scientific knowledge and for your commitment to maintaining the high standards of PLOS ONE. We greatly appreciate your support throughout this process. If you have any further questions or require additional information, please do not hesitate to contact us.

Sincerely,

---

## [Decision Letter · Decision Letter 1]

11 Mar 2024

PONE-D-23-31891R1The performance model of logistic distribution centers: Quality Function Deployment based on the Best-Worst MethodPLOS ONE

Dear Dr. Huynh,

Thank you for submitting your manuscript to PLOS ONE. After careful consideration, we feel that it has merit but does not fully meet PLOS ONE’s publication criteria as it currently stands. Therefore, we invite you to submit a revised version of the manuscript that addresses the points raised during the review process.

We look forward to receiving your revised manuscript.

Kind regards,

Jin Liu

Academic Editor

PLOS ONE

Additional Editor Comments:

Issues raised for both the original submission and for the revised paper should be fully addressed.

Reviewers' comments:

Reviewer's Responses to Questions

**Comments to the Author**

1. If the authors have adequately addressed your comments raised in a previous round of review and you feel that this manuscript is now acceptable for publication, you may indicate that here to bypass the “Comments to the Author” section, enter your conflict of interest statement in the “Confidential to Editor” section, and submit your "Accept" recommendation.

Reviewer #4: All comments have been addressed

Reviewer #5: (No Response)

Reviewer #6: All comments have been addressed

2. Is the manuscript technically sound, and do the data support the conclusions?

Reviewer #4: Yes

Reviewer #5: Partly

Reviewer #6: Partly

3. Has the statistical analysis been performed appropriately and rigorously? 

Reviewer #4: Yes

Reviewer #5: N/A

Reviewer #6: Yes

4. Have the authors made all data underlying the findings in their manuscript fully available?

Reviewer #4: Yes

Reviewer #5: Yes

Reviewer #6: Yes

5. Is the manuscript presented in an intelligible fashion and written in standard English?

Reviewer #4: Yes

Reviewer #5: No

Reviewer #6: Yes

6. Review Comments to the Author

Reviewer #4: Thanks for the opportunity to review the paper.

The reviews made are sufficient for the paper publication. I recommend the publication of the article.

Reviewer #5: Although some work has been done in this article, there are still some issues that need improvement:

(1) The research motivation of this article is not clear enough, which is why the study was conducted and why BWM was used. From my understanding, it should be an improved version of AHP. Its characteristics should be emphasized and clearly described.

(2) The contribution of this article needs to be clarified, especially in terms of theoretical research. The current form only emphasizes the applicability of BWM. So is there any theoretical contribution?

(3) The literature review in this article lacks logic and completeness, for example, some of the latest relevant literature is missing, such as research literature on BWM, DOI: 10.1016/j-eswa.2023.121227, the authors can update it.

(4) The conclusion of this article needs to be reorganized, clarifying the theoretical and applied contributions, as well as future research prospects.

(5) The format and language of this article need to be carefully revised.

Reviewer #6: • Literature review part should be extended. There are many important researches on QFD and BWM, the authors should consider these studies.

• Managerial implications and discussions should be provided.

• The authors should explain how they obtained the data, who are the experts, what are their backgrounds.

7. PLOS authors have the option to publish the peer review history of their article (what does this mean?). If published, this will include your full peer review and any attached files.

Reviewer #4: **Yes: **Diego A. de J. Pacheco

Reviewer #5: No

Reviewer #6: No

---

## [Author Response · Author response to Decision Letter 1]

3 Apr 2024

Dear Academic Editor and Reviewers,

I hope this letter finds you well. I am writing to submit the revised version of my manuscript, "The performance model of logistic distribution centers: Quality Function Deployment based on the Best-Worst Method," which was originally submitted on October 1st 2023. I appreciate the careful review of the manuscript by the reviewers and the valuable feedback provided. I would like to express my gratitude for the constructive comments and suggestions made by the reviewers. Their insights have significantly contributed to the improvement of the manuscript. In response to the reviewers' comments, we have made the following key revisions. Particulary, the red texts are responses to Reviewers.

Responses to Reviewer #4:

Thanks for the opportunity to review the paper. The reviews made are sufficient for the paper publication. I recommend the publication of the article.

Answer: Thanks so much for your suggestion. We are very grateful to your effort for reviewing our manuscript.

Responses to Reviewer #5: Although some work has been done in this article, there are still some issues that need improvement:

(1) The research motivation of this article is not clear enough, which is why the study was conducted and why BWM was used. From my understanding, it should be an improved version of AHP. Its characteristics should be emphasized and clearly described.

Answer: Thanks so much for your comment. We are very grateful to your effort for reviewing our manuscript. We would like to explain as follows:

This study was carried out from three main research motivations:

The first one is the indispensable role of Logistical Distribution Centers (LDCs) to the economic growth of nations. Accordingly, improving LDCs' performance has been receiving much attention from academics and practitioners. Still, there is little research on how to improve LDCs' performance. The first motivation is stated in Rows 40-49 in our manuscript. 

The second one is that the pertinent research in terms of boosting LDCs' operating performance often relates to customer perspectives, which can be defined as "what" issues. However, the strategies to satisfy customer needs are closely associated with service technical requirements (STRs) of LDCs, which can be defined as "how" issues. More crucially, the way to translating customer needs (i.e., CRAs) into specific engineering and design requirements (i.e., STRs) is still questionable in the relevant literature. This motivation is stated in Rows 50-62.

And the last motivation is usage of Best-Worst Method (BWM) to overcome the drawback of pairwise comparison-based methods, for instance, Analytic Hierarchy Process (AHP), Analytic network process (ANP). It is argued that BWM requires fewer comparison pairs than other pairwise comparison methods (i.e., AHP, ANP). For instance, BWM only needs pairwise comparisons, while AHP necessitates ones. Besides, BWM uses integers from 1, 2,…, and 9 to do pairwise comparisons while AHP and ANP require both integer numbers and fractional numbers. Accordingly, BWM can be easily applied in many real-world cases in terms of evaluating and prioritizing different options based on MCDA. This motivation is stated in Rows 63-78.

On top of that the characteristics of BWM were emphasized and clearly described in Rows 116-142.

(2) The contribution of this article needs to be clarified, especially in terms of theoretical research. The current form only emphasizes the applicability of BWM. So is there any theoretical contribution?

Answer: Thanks so much for your comment. We are very grateful to your effort for reviewing our manuscript. Some theoretical contributions of this paper are as follows:

First, the adoption of the SERQUAL model allows to explore and identify CRAs’ of the LDCs’ operations, which include 20 CRAs and 5 dimensions. This is also the foundation for LDCs’ operators in building service management requirements to satisfy their customers' requirements. It is illustrated that this model can be extended to further investigate CRAs for a specific operation of LDCs' performance, such as customer services, or business affairs.

Second, the utilization of the the QFD (Quality Function Deployment) model helps decision-makers in translating customer needs (i.e., CRAs) into specific engineering and design requirements (i.e., STRs). Although QFD is not new, this is the first time the “how” issues in the LDCs can be solved by the implementation of QFD. It would be said that the QDF model can be applicable in the design process and product development of various distribution systems, for instance, electric power distribution, gas distribution systems, agricultural product distribution, etc.

The last theoretical contribution is to adopt BWM to estimate the vector of priority weight of CRAs and STRs. To the best of our knowledge, this is also the first time BWM is studied to assess operations of LDCs. Thus, thanks to some strengths of BWM, as mentioned earlier, it can be efficiently used for multiple-criteria decision analysis (MCDA).

By the way, the theoretical and practical contributions of this paper are summarized in the conclusion section (Seeing Rows 442-447).

(3) The literature review in this article lacks logic and completeness, for example, some of the latest relevant literature is missing, such as research literature on BWM, DOI: 10.1016/j-eswa.2023.121227, the authors can update it.

Answer: Thanks so much for your suggestion. The suggested research is really informative. So, we referred to this paper for specific details of BWM (Seeing Rows 131-132). Besides, as requested by the another reviewer, the paper also updates some latest research on BWM and QFD (Seeing Rows 94, 101, 129).

(4) The conclusion of this article needs to be reorganized, clarifying the theoretical and applied contributions, as well as future research prospects.

Answer: Thanks so much for your suggestion. The conclusion of this article was reorganized to reflect the theoretical and applied contributions (Seeing Rows 442-447. And the future research prospects was illustrated in the last part of the paper (Seeing Rows 448-455).

(5) The format and language of this article need to be carefully revised.

Answer: Thanks so much for your suggestion. We had one English native speaker revise our manuscript to ensure that it is grammatical.

Responses to Reviewer #6: 

(1) Literature review part should be extended. There are many important researches on QFD and BWM, the authors should consider these studies.

Answer: Thanks so much for your suggestion. We revised the literature review part by updating some latest research on QFD and BWM (Seeing Rows 131-132; Rows 94, 101, 129).

(2) Managerial implications and discussions should be provided.

Answer: Thanks so much for your suggestion. The managerial implications and discussions are shown in Section 4 (Seeing Rows 331-377)

(3) The authors should explain how they obtained the data, who are the experts, what are their backgrounds.

Answer: Thanks so much for your suggestion. We would like to explain as follows:

a) The way to collect data:

According to the Vietnam Ministry of Industry and Trade (2023), there are 33 LDCs across Vietnam as of 2022. More specifically, the vast majority of these are located in the major cities and industrial provinces of Vietnam, such as HCM city, Hanoi, Danang, Dong Nai, Binh Duong, and Hai Phong. Some LDCs with a major operational scale include Gemadept, TBS Logistics, Vinalines Logistics, Saigon Newport, Transimex, DKSH Vietnam, Damco Vietnam, etc. To select a representative sample, this current study chose the four biggest LDCs, viz., Gemadept, JBS Logistics, Vinalines Logistics, and Saigon Newport for the empirical case. After that, the research team asked these LDCs to provide 10 major customers. As a result, we had a sample of 40 respondents for the survey. Because of adopting BWM to determine CRAs' weight, referring to Rezaei (2015), this paper designed the nine-point pairwise-comparison scale, as presented in Table 3, to survey 40 respondents, as mentioned above. Note that due to the highly professional nature of the questions in the questionnaire, all of the participants being surveyed were required to have adequate professional expertise in the logistics industry. From April to July 2023, we managed to gather information from such 40 experts. Nevertheless, 3 responses were discarded as they did not satisfy the consistency of BWM. Consequently, only 37 responses are valid for the next analysis.

By the way, please refer to Rows 234-248 for more information.

b) The experts and their background:

The experts in our research are customers of logistic distribution centers (LDCs). Besides, anwering questions in BWM is highly professional. Accordingly, we randomly sampled the most appropriate interviewees with the following prerequisite conditions: (1) at least 5 years of working experience in LDCs, (2) holding the position of senior staff or above, and (3) over twenty-five years of age.

Respondents' background is also shown in Table 4. Evidently, 62.2% of respondents hold managerial positions at their workplaces while 56.8% of them have working experience of more than 11 years in the distribution field. Additionally, 70.3% of respondents working at LDCs have an operating year of more than 15 years. Moreover, some main types of products of LDCs include pharmaceuticals and healthcare products, industrial and construction equipment, Fast-Moving Consumer Goods (FMCG), apparel and fashion, automotive parts and accessories, and electronics and telecommunications equipment.

By the way, please refer to Rows 251-256 and Rows 620-621 for more information.

Once again, thank you for your dedication to advancing scientific knowledge and for your commitment to maintaining the high standards of PLOS ONE. We greatly appreciate your support throughout this process. If you have any further questions or require additional information, please do not hesitate to contact us.

Sincerely,

Nguyen Tan Huynh

huynhtannguyen@dntu.edu.vn

Faculty of Economics-Management, Dong Nai Technology University, Bien Hoa City, Vietnam.

---

## [Decision Letter · Decision Letter 2]

27 May 2024

The performance model of logistic distribution centers: Quality Function Deployment based on the Best-Worst Method

PONE-D-23-31891R2

Dear Dr. Huynh,

We’re pleased to inform you that your manuscript has been judged scientifically suitable for publication and will be formally accepted for publication once it meets all outstanding technical requirements.

Kind regards,

Jin Liu

Academic Editor

PLOS ONE

Additional Editor Comments (optional):

This manuscript can be accepted now.

Reviewers' comments:

Reviewer's Responses to Questions

**Comments to the Author**

1. If the authors have adequately addressed your comments raised in a previous round of review and you feel that this manuscript is now acceptable for publication, you may indicate that here to bypass the “Comments to the Author” section, enter your conflict of interest statement in the “Confidential to Editor” section, and submit your "Accept" recommendation.

Reviewer #7: All comments have been addressed

Reviewer #8: (No Response)

2. Is the manuscript technically sound, and do the data support the conclusions?

Reviewer #7: Yes

Reviewer #8: (No Response)

3. Has the statistical analysis been performed appropriately and rigorously? 

Reviewer #7: Yes

Reviewer #8: (No Response)

4. Have the authors made all data underlying the findings in their manuscript fully available?

Reviewer #7: Yes

Reviewer #8: (No Response)

5. Is the manuscript presented in an intelligible fashion and written in standard English?

Reviewer #7: Yes

Reviewer #8: (No Response)

6. Review Comments to the Author

Reviewer #7: All comments are effectively addressed. This article has been revised to meet the standards for publication. I have no more comments.

Reviewer #8: The reviews made are sufficient for the paper publication. I recommend the publication of the article.

7. PLOS authors have the option to publish the peer review history of their article (what does this mean?). If published, this will include your full peer review and any attached files.

Reviewer #7: No

Reviewer #8: No

---

## [Editor Report · Acceptance letter]

4 Jul 2024

PONE-D-23-31891R2 

PLOS ONE

Dear Dr. Huynh, 

I'm pleased to inform you that your manuscript has been deemed suitable for publication in PLOS ONE. Congratulations! Your manuscript is now being handed over to our production team.

Kind regards, 

on behalf of

Professor Jin Liu 

Academic Editor

PLOS ONE